# High-resolution, large field-of-view label-free imaging via aberration-corrected, closed-form complex field reconstruction

**Ruizhi Cao** [1,2] ✉, **Cheng Shen** [1,2] **& Changhuei Yang** [1]

Computational imaging methods empower modern microscopes to produce high-resolution, large field-of-view, aberration-free images. Fourier ptychographic microscopy can increase the space-bandwidth product of conventional microscopy, but its iterative reconstruction methods are prone to parameter selection and tend to fail under excessive aberrations. Spatial Kramers–Kronig methods can analytically reconstruct complex fields, but is limited by aberration or providing extended resolution enhancement. Here, we present APIC, a closed-form method that weds the strengths of both methods while using only NA-matching and darkfield measurements. We establish an analytical phase retrieval framework which demonstrates the feasibility of analytically reconstructing the complex field associated with darkfield measurements. APIC can retrieve complex aberrations of an imaging system with no additional hardware and avoids iterative algorithms, requiring no human-designed convergence metrics while always obtaining a closed-form complex field solution. We experimentally demonstrate that APIC gives correct reconstruction results where Fourier ptychographic microscopy fails when constrained to the same number of measurements. APIC achieves 2.8 times faster computation using image tile size of 256 (length-wise), is robust against aberrations compared to Fourier ptychographic microscopy, and capable of addressing aberrations whose maximal phase difference exceeds 3.8π when using a NA 0.25 objective in experiment.

The pursuit of microscopy techniques that can simultaneously provide high-resolution and large field-of-view (FOV) can improve digital pathology and be broadly applied in other high-throughput imaging applications. Computational imaging, a keystone of modern microscopy, plays a crucial role in achieving such goals. Over the past few decades, remarkable progresses have been made in both fluorescence and label-free imaging field[1–7]. One such representative label-free technique, Fourier ptychographic microscopy (FPM), leverages the power of computation to provide high-resolution and aberration correction abilities to low numerical aperture (NA) objectives[1,2,8]. FPM operates by collecting a series of low-resolution images under tilted illumination and applies a core iterative phase retrieval algorithm to reconstructs sample's high spatial frequency features and optical aberration, resulting in high-resolution aberration-free imaging that preserves the inherently large FOV associated with the low numerical aperture objectives. It greatly increases the spatial bandwidth product of standard microscopy in a simple but surprisingly effective way. Due to these attractive traits, FPM has found diverse applications in quantitative phase imaging, aberration metrology, digital pathology, and other fields[2,9].

Although FPM is an important advancement in label-free microscopy, its essential iterative reconstruction algorithm poses several

[1]Department of Electrical Engineering, California Institute of Technology, Pasadena, CA, USA. [2]These authors contributed equally: Ruizhi Cao, Cheng Shen. ✉e-mail: rcao@caltech.edu

challenges. First and foremost, the iterative reconstruction of FPM is a non-convex optimization process, which means that it is not guaranteed to converge onto the actual solution[1,2,8,10–13]. In practice, the algorithm executes alternating projections between real space and spatial frequency space until certain conditions are met, such as its loss function decreasing rate reaches a lower bound, the execution reaches the allowed maximum iteration number, or the algorithm satisfies other pre-defined metric thresholds[1,2,10–15]. As a result, FPM does not guarantee that the global optimal solution is ever reached. This is problematic for exacting applications, such as digital pathology, where even small errors in the image are not tolerable. Furthermore, the joint optimization of aberration and sample spectrum can fail when the system's aberrations are sufficiently severe—leading to poor reconstructions[16]. The iterative nature of FPM reconstruction algorithm has prompted researchers to adapt machine learning concepts to its implementation, in pursuit of computational load reduction, artifact abatement, and aberration correction[17–20]. These, in turn, lead to other problems, such as contextual sensitivity and potentially greater drift away from the global optimal solution. It is worth considering at this juncture whether it is possible to develop a closed-form solution to this class of computational imaging problems, so that all these challenges can be more effectively addressed.

Recent studies have shown that the complex field can be non-iteratively reconstructed in one specific varied illumination microscopy scenario by matching the illumination angle to the objective's maximal acceptance angle (the NA-matching angle) and exploiting the signal analyticity, for example, through spatial-domain Kramers–Kronig imaging[21–23]. These findings are important and impactful as they eliminate the need for an iterative reconstruction framework and do not require a human-engineered convergence criterion. However, it is worth noting that this approach does not possess the capability to correct hybrid aberrations nor provide great resolution enhancement beyond the diffraction limit of the objective NA. As such, FPM remains a more appealing choice in various scenarios.

In this study, we present an analytical method, termed Angular Ptychographic Imaging with Closed-form method (APIC), that weds the strengths of both methods. APIC builds on complex field reconstruction using Kramers–Kronig relations and employs analytical techniques to retrieve aberration and reconstruct the darkfield associated high spatial frequency spectrum. By using NA-matching and darkfield measurements, APIC is capable of retrieving high-resolution, aberration-free complex fields when a low magnification, large FOV objective is used for data acquisition. From both simulations and experiments, APIC demonstrates unprecedented robustness against aberrations, while FPM drastically fails. Due to its analytical nature, APIC is inherently insensitive to optimization parameters and offers a guaranteed analytical complex field solution. We additionally show that APIC performs better than FPM when subjected to the same constraint on input data size, as it does not require an overly large data redundancy needed by FPM for a good convergence. By incorporating darkfield measurements, APIC effectively achieves the same theoretical resolution enhancement as FPM. We believe APIC represents an impactful step forward in the field of computational imaging.

## Results

### Principle

APIC collects both NA-matching and darkfield intensity measurements for high-resolution reconstruction. Its reconstruction process begins by analytically solving for the sample's spatial frequency spectrum and aberration with the NA-matching measurements. Then, the darkfield measurements are used to extend the sample's spatial frequency spectrum to greatly enhance the resolution of a NA-limited imaging system. The system setup, data acquisition process, and its reconstruction flowchart are illustrated in Fig. 1. In APIC's data acquisition step, the LEDs whose illumination angles match up with the maximal

acceptance angle of the imaging system are sequentially lit. The measurements under these NA-matching angle illuminations constitute the NA-matching measurements of APIC. LEDs whose illumination angles are greater than the acceptance angle are then successively lit for acquiring darkfield measurements. In the following sections, we use the word "spectrum" as shorthand for spatial frequency spectrum (the Fourier transform of the sample's complex field). We note that the spectrum is different from the Fourier transform of an acquired image, which is the Fourier transform of a pure intensity measurement.

APIC operates by first reconstructing the complex field corresponding to NA-matching measurements using Kramers–Kronig relations. These measurements are taken with LED illumination angles that match with the objective's maximal receiving angle. For a realistic imaging system, aberrations are inevitably superimposed on the spectrums' phase. To extract the objective's aberrations, we focus on the overlapping region in their spectrums (the overlap of two translated CTFs, where the translation is shown on the left side of Fig. 2). As the sample dependent phases are identical in the overlapped region of the two spectrums, subtracting their phases cancels out the sample dependent phase term, leaving only the phase differences between different parts of the pupil function (see Fig. S16 for more information). Consequently, the overlapping regions give us a linear equation with respect to the aberration term. By solving this linear equation, the aberration of the imaging system can be extracted, which can then, in turn, be used to correct the original reconstructed spectrums. The corrected spectrums are then stitched together to obtain an aberration-free, two-fold resolution-enhanced sample image.

We can then extend the spectrum by using the darkfield measurements. In this step, the reconstruction spectrum and the aberration obtained in the first step serve as the a priori knowledge. The step-by-step reconstruction operates in the following way. We choose a measurement whose spectrum is closest to the known spectrum (say, the $i$th measurement) and crop out the known spectrum based on what is sampled in this measurement, as shown in Fig. 2. This cropped spectrum, however, only contains part of the information of the $i$th measurement. Our goal is to recover the unknown part of the spectrum so that it can be filled in for spectrum spanning.

We can see that the Fourier transform of our $i$th intensity measurement consists of cross-correlation of the known and unknown spectrum and their autocorrelations. In the following, we show that by using the known spectrum, we can construct a linear equation with respect to the unknown spectrum, which can be analytically solved.

First, the autocorrelation of the known part is calculated and subtracted from the Fourier transform of the measurement. After subtraction, the autocorrelation of the unknown part and the cross-correlations are left. One important observation is that these parts are not fully coincided in the spatial frequency domain (Fig. 2). As such, we can focus on the non-overlap region where the cross-correlation solely contributes to the signal.

We can then construct a linear equation with respect to the unknown spectrum. When calculating the cross-correlation, one of the signals is shifted and multiplied with another signal. The correlation coefficient is the summation of this product. Assuming one of the two signals is known, we essentially use the known signal as the weights and calculate the summation of a weighted version of the other signal to find this coefficient. This is a linear operation. Thus, applying the known spectrum, we can construct a linear operator that takes the unknown spectrum and produces this cross-correlation. By extracting the non-overlapping part of the cross-correlation term, we can form and analytically solve a linear equation with respect to the unknown spectrum. That is, we obtain the closed-form solution of the unknown spectrum by solving this equation.

For a practical imaging system, we need to consider its aberration in the imaging process. To match up with the measurement,

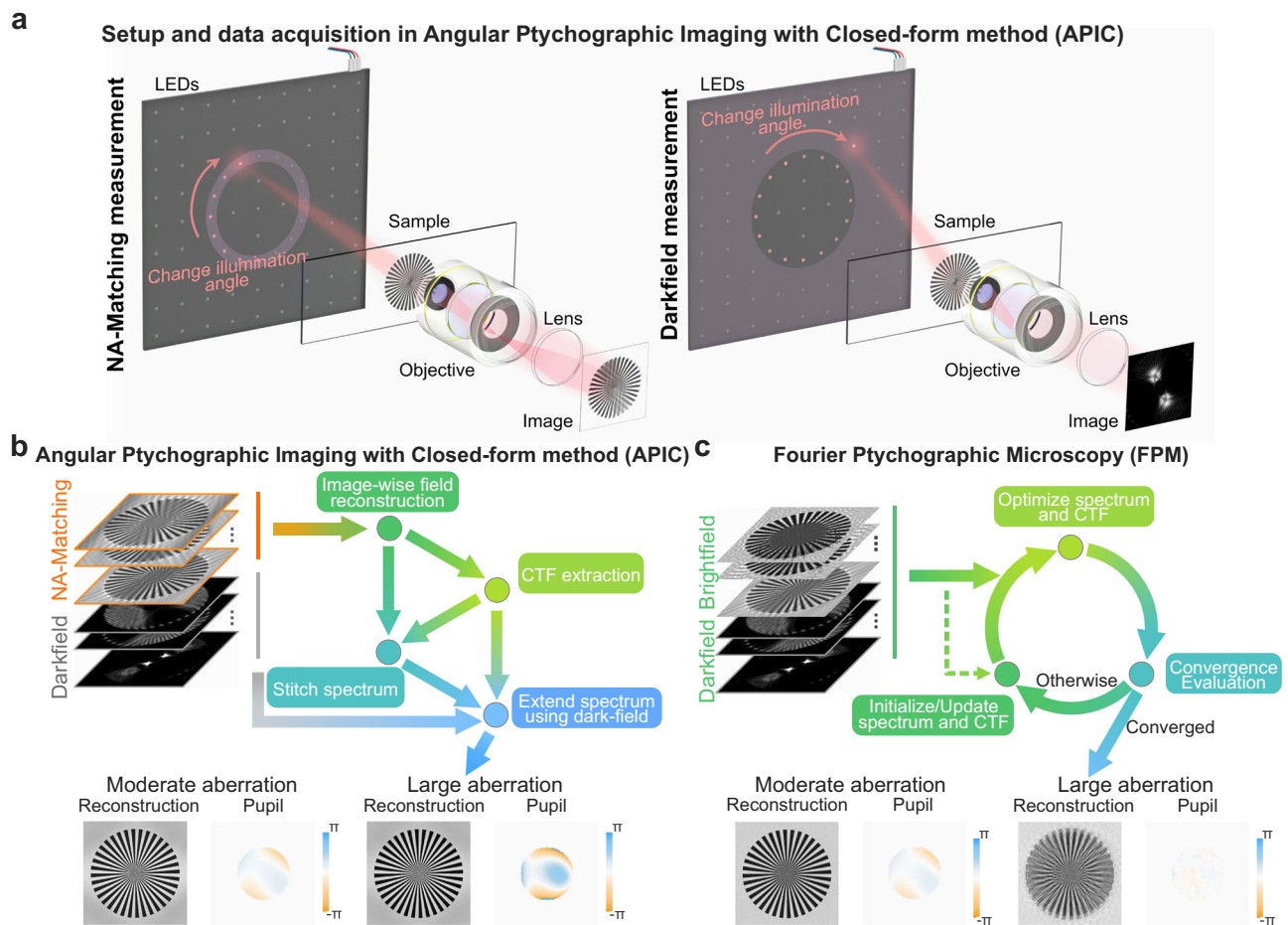

**a** Setup and data acquisition in Angular Ptychographic Imaging with Closed-form method (APIC)

**b** Angular Ptychographic Imaging with Closed-form method (APIC)

**c** Fourier Ptychographic Microscopy (FPM)

**Fig. 1 | Concept of angular ptychographic imaging with closed-form method (APIC) and comparison between the reconstruction process of APIC and Fourier ptychographic microscopy (FPM). a** Setup of APIC. The LEDs whose illumination angle matches up with the numerical aperture (NA) of the objective are lit sequentially to obtain the NA-matching measurements. Then, the LEDs whose illumination angle is larger than the objective's receiving angle are successively lit for the darkfield measurements. **b** Reconstruction process of APIC. Once the aberration is extracted, it is used to correct aberration in the image-wise field reconstruction. The aberration-corrected spectrums are then stitched together and the recovered aberration is initially introduced to the cropped out known spectrum. After recovering the unknown spectrum, the aberration gets corrected, and this corrected spectrum is filled back into the reconstructed spectrum. This process stops when all darkfield measurement has been reconstructed. The detailed derivation of the aforementioned analytical complex field reconstruction and aberration extraction methods can be found in Section 12 in the supplementary note.

serves as a prior knowledge in the spectrum extension. Using darkfield measurements, the spectrum is furtherly extended to obtain a high-resolution, aberration-free reconstruction. **c** Reconstruction process of Fourier ptychographic microscopy (FPM). FPM iteratively updates the spectrum and the aberration to minimize the differences in the measurement and reconstruction output. This iterative process is terminated upon convergence to obtain the spectrum and coherent transfer function (CTF) estimate. Pupil in the figure denotes the reconstructed aberrations.

Once the above steps are completed, we obtain a high-resolution and aberration-free sample image. The theoretical optical resolution of APIC is determined by the sum of the illumination NA and the objective NA, which is identical to FPM's NA-resolution formulae[1]. We note that FPM requires an iterative process to recover the spectrum and is sensitive to the choice of optimization parameters. On the other hand, APIC analytically recovers the actual spectrum. This direct and efficient approach sets APIC apart from FPM, offering a more straightforward and robust spectrum recovery process. In the following section, we will report on our experimental demonstration that APIC is computationally efficient and achieves consistent, high-quality complex field reconstructions even under large aberrations, whereas FPM struggles due to the increased complexity in its optimization problem.

## Experiment results

We used a low magnification objective (10× magnification, NA 0.25, Olympus) for data acquisition. A LED ring (Neopixel ring 16, Adafruit) glued onto a LED array (32 × 32 RGB LED Matrix, Adafruit) served as the illumination unit. The two LED clusters were mounted on a motorized stage for position and height adjustment, and they were individually controlled by two Arduino boards (Arduino Uno, Arduino). In the acquisition process, we lit up one LED at a time, and simultaneously triggered the camera (Prosilica GT6400, Allied Vision) to capture an image when the LED was on. This process continued until all desired LEDs were lit once. We then performed reconstruction using both APIC and FPM. The calibration of the system can be found in Section 1 in the supplementary note.

In our first experiment, we imaged a Siemens star target and chose to acquire a small dataset to perform reconstruction using APIC and FPM. The dataset acquired in this experiment consisted of 9 brightfield measurements, 8 NA-matching measurements, and 28 darkfield measurements. We note that there are works that apply multiplexed illumination scheme to reduce the number of the measurements in FPM[11,12], these methods are not as reliable as the conventional FPM data acquisition scheme. Thus, we only focus on the more reliable

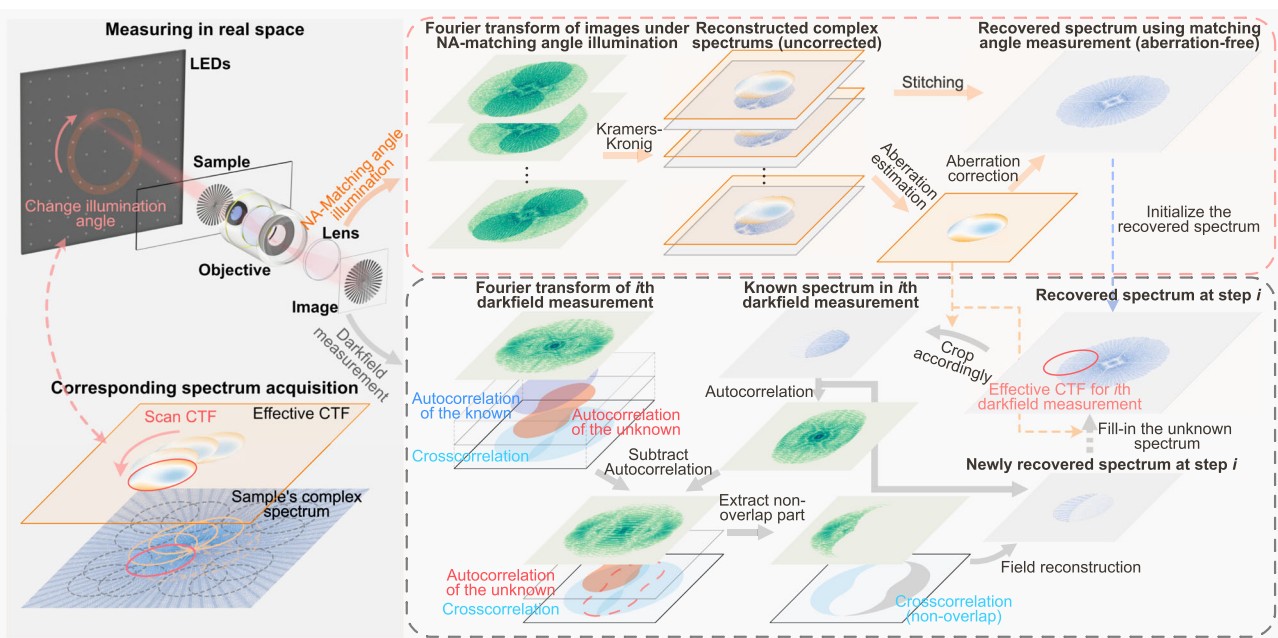

**Fig. 2 | Reconstruction pipeline for APIC.** By changing the illuminating angle, we effectively shift the CTF to different positions in the spatial frequency domain, and samples different regions of sample's spectrum. For measurements under NA-matching angle illumination, we first use Kramers–Kronig relation to recover the corresponding spectrums. The phase differences of two spectrums with overlaps in their sampled spectrum are used to extract the imaging system's aberration. Then, the image-wise reconstructed spectrums are corrected for aberration and get stitched, which forms our prior knowledge in the reconstruction process involving darkfield measurements. To extend the spectrum using darkfield measurement, the known spectrum in the $i$th measurement is used to isolate cross-correlation from other autocorrelation terms. By solving a linear equation involving the isolated cross-correlation, the unknown spectrum can be analytically obtained. We then use the newly reconstructed spectrum to extend the recovered spectrum. The extended spectrum then serves as the prior for the $(i+1)$th measurement.

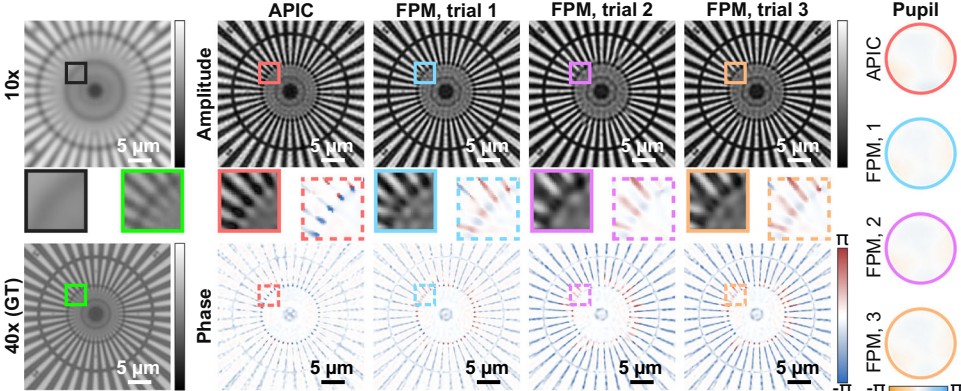

**Fig. 3 | Reconstruction result of APIC and FPM using a small number of measurements.** For comparison, we also acquired a ground truth (GT) image, which was imaged under a high-NA, 40× objective. The Kohler illuminated image under the same 10× objective used in APIC's data acquisition is shown on the upper left. The reconstructed pupils are shown on the right side of the figure. For FPM reconstructions, we selected three representative results from all 6 parameter sets we used in FPM, and trial 1 is the best result we got. We note that when the ground truth is unknown, all FPM reconstruction results might be falsely treated as the correct solution as they all possess good contrast and fine details. However, apparent discrepancies are noticeable when comparing these results with the ground truth image. APIC, as an analytical method, is not prone to parameter selection.

acquisition scheme in this study. The nominal scanning pupil overlap rate was approximately 65%. In our experiments, the second-order Gauss-Newton FPM reconstruction algorithm was applied for reconstruction as it was found to be the most robust FPM reconstruction algorithm[13], and the sum of all measurements was used for initialization. We also note that we used 6 sets of parameters in the reconstruction of FPM and chose the best result, as the reconstruction quality of FPM heavily depends on its parameters. Some representative FPM results are also shown in Fig. 3, which confirms such parameter dependency. On the contrary, the faithfulness and correctness are guaranteed in APIC, benefiting from its analytical phase retrieval framework. We found that APIC was able to render the correct complex field while FPM failed, as shown in Fig. 3. As shown by the result, the reconstructed finer spokes were distorted in all reconstruction results of FPM. Moreover, noticeable wavy reconstruction artifacts existed in the phases reconstructed by FPM. When the measurements were given to APIC, the reconstructed phases and amplitudes were less wavy. The reconstructed amplitude is also closer to the ground truth, which is

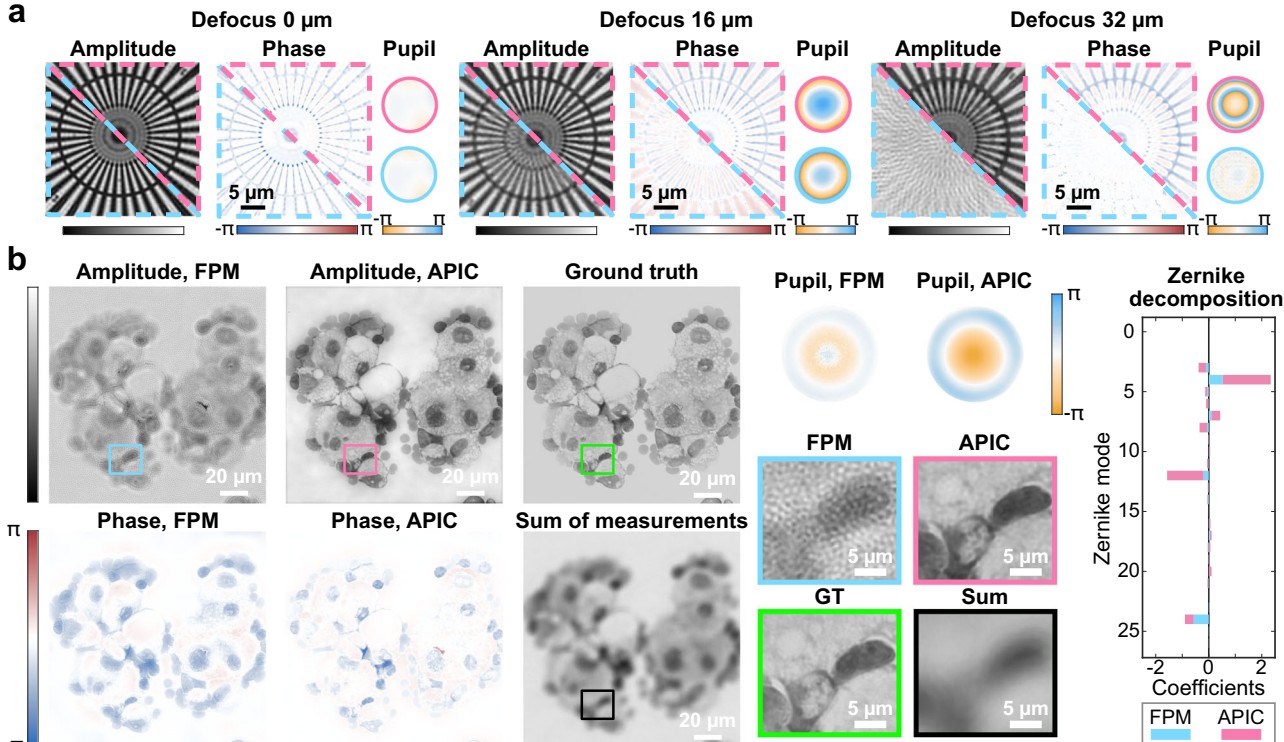

**Fig. 4 | Reconstruction under different levels of aberrations. a** Reconstructed complex fields and aberrations with different defocus distances. For the reconstruction of APIC and FPM, the defocus distance is labeled on top of each group. In our reconstruction, the actual defocus distance is hidden from both algorithms. APIC reconstructed amplitude and phase are shown on the upper right of each group and highlighted by the dashed magenta line. FPM reconstructed amplitude and phase are shown on the lower left of each group and highlighted by the dashed cyan line. The reconstructed pupils of APIC and FPM are also color-coded by magenta and cyan, respectively. **b** Reconstruction of a human thyroid carcinoma cell sample using APIC and FPM. Sum of measurements denotes the summation of all 316 images we acquired, which can be treated as the incoherent image we would get under the same objective. The ground truth was acquired using an objective whose magnification power is 40, and NA equals 0.75. The NA of this 40× objective equals the theoretical synthetic NA of APIC and FPM. We calculated the square root of the summed image and the 40× image to match up with the amplitude reconstruction. The zoomed images of the highlighted boxes are shown on the lower right of **b**. The Zernike decompositions of retrieved aberrations using FPM and APIC are shown on the far-right side of **b**.

sampled using a high-NA objective. We stress that when the ground truth is not given, all three FPM results shown in Fig. 3 may be perceived as a good reconstruction in practice as they preserve good contrast and are detail-rich. However, these reconstructions are of low fidelity as they all deviate from the ground truth we acquired. We can find some of the spokes of the Siemens star target were missing in the FPM's reconstruction, which indicates the failure of FPM. This experiment showcased the ability of APIC to better retrieve a high-resolution complex field when the raw data size is constrained because it is an analytical method and does not rely as heavily on pupil overlap redundancy for solution convergence that FPM requires.

It is also worth noting, for input image tile of length 256 pixels on both sides, APIC reconstruction took 9 seconds on a personal computer (CPU: Intel Core i9-10940X with 64 GB RAM), while FPM required 25 seconds to finish the reconstruction. The relative computational efficiency of APIC can again be attributed to the analytical nature of its approach in contrast to FPM. We note that this computational efficiency is image tile size dependent—the smaller the tile is, the more efficient APIC can be (see section 6 in the supplementary note for more information). As it is generally preferred to divide a large image into smaller tiles in parallel computing, APIC's computational efficiency for smaller tiles aligns well with practical computation considerations.

In the next experiment, we studied the robustness of APIC and FPM in addressing optical aberrations. For this experiment, we acquired a total of 316 images, which consisted of 52 normal brightfield measurements, 16 NA-matching measurements, and 248 darkfield measurements. The nominal scanning pupil overlap ratio of our

dataset was ~87%, and the final theoretical synthetic NA was equal to 0.75 when all darkfield measurements were used. We note that this large degree of spectrum overlap was chosen to provide sufficient data redundancy for the best performance of FPM. APIC does not require such a large dataset (examples can be found in Fig. 3 and Fig. S3 of our supplementary note). In our reconstruction, APIC only used the NA-matching and darkfield measurements, whereas FPM used the entire dataset, including these additional 52 brightfield measurements corresponding to illumination angles that were below the objective's acceptance angle.

We deliberately defocused a Siemens star target to assess how the two methods perform under different aberration levels. In this experiment, the sample was defocused to different levels, and the defocus information was hidden from both methods. The results of FPM and APIC are shown in Fig. 4a. Clearly, for large aberrations whose phase standard deviation exceeded 1.1π (the case when Siemens star target was defocused by 32 μm, and the maximal phase difference is -3.8π), FPM failed to find the correct solution and the reconstructed images were considerably different from the ground truth, even when the algorithm indicated its convergence criterion was reached. At a lower aberration level, the amplitude reconstructions of FPM appeared to be close to the ideal case. However, the reconstructed phases were substantially different from the result when no defocus was introduced. In contrast, APIC was highly robust to different levels of aberrations. Although the contrast of APIC's reconstruction dropped under larger aberrations, it retrieved the correct aberrations and gave high-resolution complex field reconstructions that matched with the in-

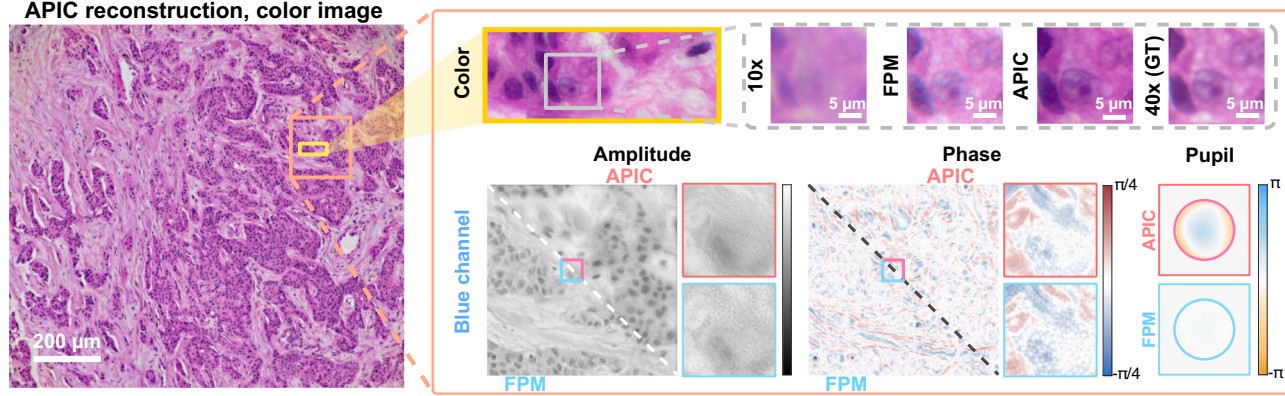

**Fig. 5 | Reconstructed high-resolution image of hematoxylin and eosin (H&E) stained breast cancer cells.** APIC reconstructed aberration corrected, high-resolution color image is shown on the left. The zoomed image of the highlighted region in the color image is shown on the right. The image with the label "10×" denotes the image acquired using the same 10× magnification objective, which was used for data acquisition and thus showed poorer resolution due to its limited NA. The color image with label "40× (GT)" denotes the ground truth we acquired using a 40× objective whose NA equals the theoretical synthetic NA of APIC. We note that

we manually focused the image under red, green, and blue LED illumination when acquiring the ground truth as the best focal planes for them are different, while no tuning was applied in APIC's data acquisition. We picked a blue channel as an example in this illustration, and the complex field reconstructions and retrieved aberrations are shown at the bottom of the rounded box. From the zoomed images, APIC shows good correspondence with the ground truth, while FPM is much noisier.

focus result. The measured resolution for both FPM and APIC is ~870 nm when the in-focus measurements were used, which is close to the 840 nm theoretical resolution (Fig. S4).

To test the two methods under more complex aberrations, we used an obsolete Olympus objective (10× magnification, NA 0.25) that was designed to work with another type of tube lens for image measurement in this particular experiment. A human thyroid adenocarcinoma cell sample was imaged to see their performance. As the standard deviation of the phase of the imaging system's aberration was close to 2π/5, FPM failed to reconstruct a high-resolution image. From Fig. 4b, the reconstructed amplitude of FPM was heavily distorted by the reconstruction artifacts. APIC recovered all the finer details that were in good correspondence with the image we acquired using a 0.75 NA objective.

We then conducted an experiment using a hematoxylin and eosin (H&E) stained breast cancer cell sample. We used red, green, and blue LEDs to acquire datasets for these three different channels and then applied APIC for the reconstruction. In this experiment, the sample was placed at a fixed height in the data acquisition process. As a result, we see different levels of defocus in different channels lying on top of the chromatic aberrations of the objective (Fig. S5). To acquire the ground truth image, we switched to a 40× objective and manually focused each channel. We calibrated the illumination angles for the central patch (side length: 512 pixels) and then calculated the angles for off-axis patches using geometry. These calibrated illumination angles were used as the input parameter in our reconstruction. The final reconstructed region is a square of side length of 1.2 mm in this experiment.

The reconstructed color image is shown in Fig. 5. The comparison of all three channels can be found in our supplementary note (Fig. S5). From the zoomed images in Fig. 5, we can see that the reconstructions of FPM were noisy for the blue channel. We found that FPM did not work well with this weakly absorptive sample under a relatively high aberration level. It failed to extract the aberration of the imaging system. As such, the color image generated by FPM appeared grainy, and the high spatial frequency information was only partially recovered. We also see that the color reconstruction of APIC retained all high spatial frequency features that were closely matched up with the ground truth we acquired. This demonstrates that the aberration and complex field reconstruction of APIC is considerably more accurate compared with FPM.

## Discussion

We showed that APIC can extract large aberrations and synthesize large FOV and high-resolution images using low NA objectives. APIC empowers computational label-free microscopy with high robustness against aberration. Under the same high-aberration conditions, FPM fails to recover the aberration, and its reconstruction result largely inherits such aberration and thus cannot produce aberration-free, high-resolution reconstructions.

Moreover, some of the fundamental problems in the conventional phase retrieval algorithm, such as being prone to optimization parameters and getting stuck in local minimum, are solved in APIC. Previous results demonstrated that reconstruction artifacts appear in FPM without properly selected parameters or loss functions[10,13,24], which is in consistent with our experiment results shown in Fig. 3. Without a properly engineered metric, the selection of parameters becomes highly subjective. This again indicates that it is often unclear on whether FPM is even converging close to the real complex field solution. APIC is robust against this problem as it does not require an iterative algorithm for reconstruction. It circumvents the need to choose optimization parameters or designing metric for convergence. However, as an analytical method, the knowledge about the position of the LEDs, as well as the alignment of the NA-matching angle illuminations, is important in APIC. When large calibration errors show up, the solution of APIC will be negatively impacted (Fig. S14). Thus, a good calibration is required in APIC to get the correct solution. In addition, we note that the amplitude of the CTF is assumed to be unit in our prototype as the aperture of our experiment system has negligible amplitude variation. For an aperture with intrinsic amplitude variation, we anticipate that this can be corrected using a similar approach applied for the aberration correction. Instead of subtracting the phase, one would calculate the ratio of the spectrum's amplitude for the overlapping part and then use this ratio to correct for the unevenness in CTF's amplitude.

As APIC directly solves the complex field, it avoids the potentially time-consuming iterative process. When a reasonable image patch size is chosen, APIC is computationally more efficient compared with FPM. As such, APIC alleviates the lengthy processing in FPM, making it a more appealing method (Fig. S7 in supplementary note).

While this work demonstrates a working APIC prototype, we note that a key aspect of the prototype would need further design improvements if a larger field of view is desired. Specifically, in this prototype, we treat the LED illumination as a plane wave at the sample

plane. However, for a large field of views, the illumination angle can be quite different for different patches in the entire FOV. This may lead to noisy reconstruction of NA-matching measurements, as a previous study has indicated[23]. We anticipate that this problem can be mitigated by increasing the distance between the LED and the sample. It can also be solved by designing better LED illumination systems. Additionally, the number of LED illuminations can be reduced in future systems by decreasing the overlap of two measured spectrums.

In conclusion, we demonstrate that APIC can provide high-resolution and large FOV label-free imaging with unprecedented robustness to aberrations. As an analytical method, APIC is insensitive to parameter selections and can compute the correct imaging field without getting trapped in local minimums. APIC's analyticity is particularly important in a range of exacting applications, such as digital pathology, where even minor errors are not tolerable. APIC guarantees the correct solution, while FPM-like iterative methods cannot. Additionally, APIC brings new possibilities to label-free computational microscopy as it affords greater freedom in the use of engineered pupils for various imaging purposes. We anticipate the APIC concept can be fruitfully adopted for other methods, such as the aberration found by APIC, which can potentially be used to correct incoherent imaging. The idea of using the known spectrum to reconstruct the unknown spectrum can be readily adapted for use in other scenarios.

### Reporting summary
Further information on research design is available in the Nature Portfolio Reporting Summary linked to this article.

## Data availability
Part of the data that supports this study is available on GitHub (https://github.com/rzcao/APIC-analytical-complex-field-reconstruction). The complete data that support the plots within this paper and other findings of this study are available from the corresponding authors upon request. Source data are provided with this paper.

## Code availability
The reconstruction code that supports the plots within this paper and other findings of this study is available on Supplemetary Code 1 as well as GitHub (https://github.com/rzcao/APIC-analytical-complex-field-reconstruction) and the zenodo repository[25].

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

## Acknowledgements
This research is supported by Heritage Medical Research Institute (HMRI) (Award number HMRI-15-09-01). The authors thank Dr. Jerome Mertz for the insightful discussion of this work.

## Author contributions
R.C. conceived the idea. R.C. conducted the theoretical analysis, conducted the simulations and wrote the reconstruction algorithm. C.S. built the experiment setup, wrote the hardware controlling code and performed the calibration of the system. R.C. and C.S. conducted the experiments. C.Y. supervised this project. All authors contributed to the preparation of the manuscript.

## Competing interests
The authors (R.C., C.S., and C.Y.) declare the following competing interests: On 30 March 2023, the California Institute of Technology filed a provisional patent application for APIC, which covered the concept and implementation of the APIC system described here.
