## [Peer Review File · Nature Communications]

High-resolution, large field-of-view label-free imaging via
aberration-corrected, closed-form complex field
reconstructionREVIEWER COMMENTS

Reviewer #1 (Remarks to the Author):

This manuscript presents an interesting alternative to the Fourier ptychographic reconstruction. The proposed analytical method for phase retrieval uses the Kramers-kroning method to reconstruct the complex spectrum and circumvents the aberration recovery issue in this method by using sequential illumination of the NA-matched LEDs in the system to solve for the unknown pupil using the redundant measurements. The darkfield reconstruction also exploits this sequential illumination process but uses a different approach since the complex spectrum is not readily available for these LEDs. Here the prior information from brightfield area is used to determine the unknown areas from the darkfield with the help of autocorrelation/cross-correlation between the known and unknown areas. Manuscript is very well written with coherent structure and illustrative figures.

This new approach is well supported by various results presented throughout the manuscript. Most scenarios that a reader would be interested are covered either in the main manuscript or in the supplementary material. Particularly, the results with strong aberrations, low SNR and fewer number of LEDs show the potential of this technique.

There are some potential questions about this technique such as the dependency on the calibration of LED positions, brightness variation between the LEDs etc., which also have been addressed by the authors in the supplementary material. This, however, suggests that several advancements are required for this new approach to be adapted more widely or to replace the existing FPM reconstruction.

Some suggestions for improving the manuscript:

- Since phase recovery is a critical component of FPM and is sought out for transparent samples' imaging, such as cell cultures, it would be helpful to have some results with phase only object like beads or cell cultures for quantitative phase evaluation.
- In simulations, the impact of different amount of aberrations have been demonstrated where all the aberrations are phase only aberrations. However, there might be scenarios

where amplitude aberrations, e.g., vignetting of the aperture, polygonal aperture etc., can be present. It would be helpful to understand how this method behaves there with some simulations.

- Photo of the experimental setup in the supplementary material would be helpful along with indication of which LEDs are taken from the matrix array since it is not clear how brightfield LEDs are selected and if the ring array mounting is blocking any darkfield LEDs etc.

- It might be helpful to add a short discussion regarding the potential for extrapolating APIC to the lensless ptychography approach.

- A process to match the ring LED illumination to the objective NA has been described in the supplementary document. How critical is this alignment and how would misalignment here would impact the krammers-koning reconstruction?

Reviewer #2 (Remarks to the Author):

Please see the attached report in PDF.

[**Editorial Note:** please see the next page for Reviewer #2's report.]

This manuscript describes a new form of computational phase microscopy technique, termed APIC, which features wide field of view (FOV), high resolution (beyond the 2NA incoherent limit), and computational aberration compensation. As compared to the current state-of-the-art, such imaging capability previously could only be achieved using the Fourier ptychographic microscopy (FPM) method, APIC provides an alternative solution. The main advantage of APIC over FPM is that APIC relies on analytic solution, bypassing the need for solving a non-convex optimization problem using iterative algorithms in FPM. Overall, I found the work is very interesting. I believe it is an impactful contribution to the field of computational imaging.

I have the following comments / suggestions:

- A) I think the following areas can be improved for the clarity of the description of the techniques and the presentation of the results:
- The (light color themed) colormap used to display the phase maps are a bit hard to see the details.
 - In all the results, the colorbar of phase maps is in the range of $[-\pi, \pi]$. This begs the following questions:
 - 1) is there any phase unwrapping issues in the final results?
 - 2) What is the meaning of the negative values? Are they reconstruction artifacts? (I believe all the samples used should only have positive phase physically).
 - 3) following 2), the negative components become quite strong in the H&E slide results. (the positive red / negative blue components seem to show equally frequent). I would like to understand a bit better on why this is the case.
 - 4) Following 3) In some other work (e.g. <https://www.ncbi.nlm.nih.gov/pmc/articles/PMC10026583/>), the phase map shows resemblance to the absorption / amplitude map. I'm a bit puzzled by what I am seeing here (again, it might be due to the confusing colormap).
- B) Questions regarding the APIC technique:
- APIC relies solely on an analytical solution, so it requires a precious knowledge of the illumination angles. In addition, with the NA-matching requirement for the brightfield (BF) measurement, the BF LEDs must be placed conjugate to the edge of the microscope objective. Based on these requirements, I have the following questions that I hope the authors can provide more details
 - 1) how strict is the NA-matching condition is? Perhaps the author can quantify the effect by the amount of NA mismatch.
 - 2) In the reviewer's own experience, when an LED approaches the NA-matching condition, significant portion of the raw intensity measurement will suffer from the vignetting artifacts – shown as a portion of the intensity measurement becomes darkfield (DF). This effect is also well-documented in various work in the context of FPM. Fundamentally this is because the LED illumination is spherical that covers a small range of NA variations within the FOV. At the critical point where the central illumination angle of the LED matches the NA, the rays that have slightly higher illumination angle will result in DF measurement. How this vignetting effect handled in APIC?
 - 3) Relating to 1) & 2), since LED illumination is spherical and finite amount of NA-matching condition tolerance, how large is the FOV one can achieve in practice? A short discussion has been given in the Discussion section. Some quantitative numbers are missing in the text.
 - As angle calibration is a critical component of APIC, I suggest to make it clear in the main text (now I can only find this discussion in the supplement) and reference the work that the calibration algorithm is based on.

C) Reference to related work

- The work has been mainly placed in the context of FPM. Due to the requirement of NA-matching measurement, it has been also placed in the context of a very recent (2021) work on space-domain Kramers–Kronig (KK) relation. I would like to make a few points:
 - The NA-matching requirement for efficient phase recovery based on intensity-only measurement was first pointed out by the work in 2019: <https://www.spiedigitallibrary.org/journals/advanced-photonics/volume-1/issue-06/066004/High-speed-in-vitro-intensity-diffraction-tomography/10.1117/1.AP.1.6.066004.full?SSO=1>
 - The 2019 work derives this critical requirement based on a transfer function (1st Born approximation) approach.
 - In a more recent work, the KK relation has been shown to be equivalent to the transfer function analysis (more precisely, an expanded 1st Rytov approximation model introduced in <https://www.nature.com/articles/s41377-022-00815-7>

Minor comments:

- What is the initialization method used in the FPM reconstruction? Has the authors tried the differential phase contrast (DPC) based initialization method in Ref. 12 to alleviate some of the artifacts seen in the results?
- “Spatial-bandwidth product” – I believe usually it refers to as “Space-bandwidth product”.
- Around Eqn S9, I suggest define $[NA/\lambda]$ to match the unit.

We appreciate the comments from the reviewers and their recognition of the significance and novelty of APIC. We have revised our manuscript based on the reviewers' suggestions and the point-by-point response is listed in this response letter.

Reviewer 1:

This manuscript presents an interesting alternative to the Fourier ptychographic reconstruction. The proposed analytical method for phase retrieval uses the Kramers-kronig method to reconstruct the complex spectrum and circumvents the aberration recovery issue in this method by using sequential illumination of the NA-matched LEDs in the system to solve for the unknown pupil using the redundant measurements. The darkfield reconstruction also exploits this sequential illumination process but uses a different approach since the complex spectrum is not readily available for these LEDs. Here the prior information from brightfield area is used to determine the unknown areas from the darkfield with the help of autocorrelation/cross-correlation between the known and unknown areas. Manuscript is very well written with coherent structure and illustrative figures.

This new approach is well supported by various results presented throughout the manuscript. Most scenarios that a reader would be interested are covered either in the main manuscript or in the supplementary material. Particularly, the results with strong aberrations, low SNR and fewer number of LEDs show the potential of this technique.

There are some potential questions about this technique such as the dependency on the calibration of LED positions, brightness variation between the LEDs etc., which also have been addressed by the authors in the supplementary material. This, however, suggests that several advancements are required for this new approach to be adapted more widely or to replace the existing FPM reconstruction.

Author response: We appreciate the reviewer for the summary of APIC and the positive comments with respect to our method. We have revised our manuscript based on the reviewer's comments.

Some suggestions for improving the manuscript:

- Since phase recovery is a critical component of FPM and is sought out for transparent samples' imaging, such as cell cultures, it would be helpful to have some results with phase only object like beads or cell cultures for quantitative phase evaluation.

Author response: We thank the reviewer for the suggestion. We can expect that since APIC recover phase through an analytical approach, it should outperform FPM in phase reconstruction. We have previously shown, in an earlier paper (Fig. 9 of Ref 23 in our main manuscript), that analytic Kramers-Kronig phase reconstruction is more accurate than the iterative phase retrieval approach. APIC shares a similar analytical root as Ref. 23.

To specifically address the reviewer's point, we have now included an experiment with phase objects as the target in Section 5 of the revised supplementary note. The sample we used in the experiment are

polystyrene beads with a nominal diameter of 3 microns and the results are shown below in Fig. R1. From the results, we can see the phase profile from APIC is closer to a semi ellipse (reflecting the bead's profile) while the result of FPM shows a tiny peak on the top of the profile which makes it deviate from the elliptical profile.

Figure R1. Experiment results using polystyrene beads. The inset figures show the line profile of the phase reconstruction result. The green dashed line in the inset of the phase reconstruction result indicates the ideal elliptical phase profile of a sphere.

- In simulations, the impact of different amount of aberrations have been demonstrated where all the aberrations are phase only aberrations. However, there might be scenarios where amplitude aberrations, e.g., vignetting of the aperture, polygonal aperture etc., can be present. It would be helpful to understand how this method behaves there with some simulations.

Author response: We thank the reviewer for the question. For our darkfield reconstruction part, we extract a linear cross-correlation term of the unknown spectrum and the reconstructed spectrum. This is the place where the shape of the aperture matters. We note that resultant shape of the correlation of two complex objects is nontrivial. A simple example is shown below:

Figure R2. Shape of the cross-correlations of different objects. The only difference between B and C is that C has one extra tiny dot on the left side of the big disk.

Although there is only a small difference between object B and object C in Fig. R1, the shape of the output cross-correlation is considerably different. More importantly, the total area of the non-zero part in the corresponding cross-correlation differs substantially. The ratio of the covered area of $A * C$ over $A * B$ is not linearly dependent on the ratio of C over B (which is close to 1). This is important as we

need to construct a linear equation from the non-overlapping region and the rank of the correlation operator depends on the area of the non-overlapping part.

Thus, it would be beneficial if to determine the shape of the aperture before the actual measurement and use it as a prior knowledge in the reconstruction code. For aperture with intrinsic amplitude variation, we anticipate that this can be corrected using a similar approach applied for the aberration correction. Instead of subtracting the phase, one would calculate the ratio of the spectrum's amplitude for the overlapping part and then uses this ratio to correct for unevenness in CTF's amplitude. This approach is not implemented in our study as we do not have this issue for the prototype of APIC, but we anticipate that it can be integrated in our future development when amplitude becomes an issue. We have revised the manuscript to clarify this point in the second paragraph of Section 4 of the main manuscript and the paragraph above Fig. S19 in our revised supplementary note.

- Photo of the experimental setup in the supplementary material would be helpful along with indication of which LEDs are taken from the matrix array since it is not clear how brightfield LEDs are selected and if the ring array mounting is blocking any darkfield LEDs etc.

Author response: Yes, the ring LED board blocks some of the LEDs. As the LED array we used in the experiment is a dense one and the spectrum overlap of two adjacent LEDs is over 85%, the measurements are more than enough for both FPM and APIC. One thing we would like to note is that the illumination angle of the LEDs being blocked by the ring is close to the maximal acceptance angle of the system, the maximal resolution we can achieved using APIC is not affected by it. With a customized LED board, we believe this issue can be avoided.

With respect to the LED selection, the full dataset uses all LEDs on the board excluding those being blocked by the ring attached onto it (see Fig. S1 for the illumination k vectors). For the reduced dataset, we sampled from the brightfield so that they are more uniform in the spatial frequency space (we first uniformly sampled from the continuous spatial frequency space and then find the nearest LED for the desired k vector). The arrangement of the LEDs can be visualized using the data we provided. Figure R3 shows the arrangement and it has been included in the supplementary (Fig. S3 in our revision).

Figure R3. **a**, Image of the ring LED on top of a LED array. A black tape is covered on the LED to prevent stray light goes into the system. **b**, Illumination k vector for the full and reduced dataset. The ring LEDs sit in between the two black circles in the figure. We note the reduced dataset is a subset of the full set. To form the reduced set, the LEDs are chosen so that they are more uniformly distributed in k-space.

The system of APIC looks just like a conventional FPM system. Although the LED illumination unit consists of two different LED boards in our prototype, the appearance of the physical system is perceptually identical to an FPM system.

We have revised the manuscript to clarify this point in Section 2 of our supplementary note.

- It might be helpful to add a short discussion regarding the potential for extrapolating APIC to the lensless ptychography approach.

Author response: We thank the reviewer for the comment. For lensless ptychography, it is difficult to define the maximal acceptance angle of the system and thus the Kramers-Kronig method is not a good fit for that modality. It is possible to adapt part of APIC such as the darkfield reconstruction for some of the lensless systems but we do not think APIC is a good fit for a general lensless system.

- A process to match the ring LED illumination to the objective NA has been described in the supplementary document. How critical is this alignment and how would misalignment here would impact the krammers-koning reconstruction?

Author response: We thank the reviewer for this important question. Indeed, APIC relies on NA-matching measurements and, as such, alignment is important for APIC. If some of the NA-matching measurements becomes darkfield measurements due to misalignment, APIC computation may not compute correctly at all, as the change would violate our assumption that the unaltered part of the illumination light enters the system. This problem can be easily avoided as darkfield measurements should appear “dark” or low in overall light signal. By using this as a guide, it is possible to calibrate the position of the LED as one could visually determine when the illumination angle matches up with the imaging system’s maximal acceptance angle by using the brightness of the measurement as a guidance.

For illumination angle that tends to be smaller than the desired angle, the reconstruction degrades with respect to the angle. We showed this degradation in our previous work (Ref 23 in our main manuscript) and the result is reproduced here:

Figure R4. Kramers-Kronig reconstruction accuracy vs inaccurate angle. Here the negative value means the k vector of the illumination light falls outside of the CTF. MSE: mean squared error; FSIM: feature similarity index.

As shown in the results, the accuracy of the reconstruction degrades gradually with respect to the illumination angle error when the illumination light passes through the system. Thus, APIC is tolerant to mismatch in the angle provided the actual illumination angle is smaller than the desired angle.

We have revised the manuscript to elaborate on this point in Section 10 of our revised supplementary note.

Reviewer 2:

This manuscript describes a new form of computational phase microscopy technique, termed APIC, which features wide field of view (FOV), high resolution (beyond the 2NA incoherent limit), and computational aberration compensation. As compared to the current state-of-the-art, such imaging capability previously could only be achieved using the Fourier ptychographic microscopy (FPM) method, APIC provides an alternative solution. The main advantage of APIC over FPM is that APIC relies on analytic solution, bypassing the need for solving a non-convex optimization problem using iterative algorithms in FPM. Overall, I found the work is very interesting. I believe it is an impactful contribution to the field of computational imaging.

Author response: We appreciate the reviewer for the accurate summary and positive feedback for APIC. We have revised our manuscript based on the reviewer's suggestions.

I have the following comments / suggestions:

A) I think the following areas can be improved for the clarity of the description of the techniques and the presentation of the results:

- o The (light color themed) colormap used to display the phase maps are a bit hard to see the details.

- o In all the results, the colorbar of phase maps is in the range of $[-\pi, \pi]$. This begs the following questions:

- o 1) is there any phase unwrapping issues in the final results?

- o 2) What is the meaning of the negative values? Are they reconstruction artifacts?

- (I believe all the samples used should only have positive phase physically).

- o 3) following 2), the negative components become quite strong in the H&E slide results. (the positive red / negative blue components seem to show equally frequent). I would like to understand a bit better on why this is the case.

o 4)Following 3)In some other work(e.g. <https://www.ncbi.nlm.nih.gov/pmc/articles/PMC10026583/>), the phase map shows resemblance to the absorption / amplitude map. I'm a bit puzzled by what I am seeing here (again, it might be due to the confusing colormap).

Author response: We thank the reviewer for providing his comments on the colormap. We note that the range of the colormap is chosen so that we do not saturate FPM reconstruction results, which looks unnatural when saturated. For the final phase result, we do need to unwrap it, just as we would do in an FPM system.

As for the negative values, we would like to kindly chose to define phase in the range of $[-\pi, \pi]$ rather than $[0, 2\pi]$. The negative values are not artifacts. The reconstructed phase of APIC is with reference to the unaltered light, which has zero phase in the reconstruction. When the sample has a lower refractive index compared with its surrounding medium, its pathlength is shorter than that of the unaltered light and thus shows a negative phase. As such, what we have shown in the manuscript are indeed features of the sample. As the raw data is provided, readers may freely choose the colormap for the visualization if they have their own preferences and match up with their empirical observations.

B) Questions regarding the APIC technique:

o APIC relies solely on an analytical solution, so it requires a precious knowledge of the illumination angles. In addition, with the NA-matching requirement for the brightfield (BF) measurement, the BF LEDs must be placed conjugate to the edge of the microscope objective. Based on these requirements, I have the following questions that I hope the authors can provide more details

o 1) how strict is the NA-matching condition is? Perhaps the author can quantify the effect by the amount of NA mismatch.

Author response: We thank the reviewer for the question. In APIC imaging, we prefer to avoid the case where the NA-matching measurements actually become darkfield measurements. As this arrangement is very prone to misalignment problems. If some of the NA-matching measurements becomes darkfield measurements due to misalignment, it would violate our assumption that the unaltered part of the illumination light enters the system. In turn, this will disrupt our APIC computation.

For illumination angle that tends to be smaller than the desired angle, the reconstruction result of Kramers-Kronig degrades with respect to the angle mismatch. We showed this degradation in our previous work (Ref 23 in our main manuscript). The result is copied below:

Figure R5. Kramers-Kronig reconstruction accuracy vs inaccurate angle. Here the negative value means the k vector of the illumination light falls outside of the CTF. MSE: mean squared error; FSIM: feature similarity index.

We have revised the manuscript to clarify this point in Section 10 of our revised supplementary note.

o 2) In the reviewer's own experience, when an LED approaches the NA-matching condition, significant portion of the raw intensity measurement will suffer from the vignetting artifacts – shown as a portion of the intensity measurement becomes darkfield (DF). This effect is also well-documented in various work in the context of FPM. Fundamentally this is because the LED illumination is spherical that covers a small range of NA variations within the FOV. At the critical point where the central illumination angle of the LED matches the NA, the rays that have slightly higher illumination angle will result in DF measurement. How this vignetting effect handled in APIC?

Author response: Thanks for raising this point as this is important to APIC. As we have explained above, the Kramers-Kronig reconstruction results degrades slowly when the illumination angle falls below the NA-matching angle. As such, we deliberately increased the distance between the LED and the sample to ensure that the region we wanted to reconstruct stayed away from darkfield when reconstructing a large area, such as the case for Fig. 5 in our main manuscript.

o 3) Relating to 1)&2), since LED illumination is spherical and finite amount of NA-matching condition tolerance, how large is the FOV one can achieve in practice? A short discussion has been given in the Discussion section. Some quantitative numbers are missing in the text.

Author response: We thank the reviewer for pointing out this missing information. Although the number can be estimated from the scale bar of Fig. 5, it is hidden in the figure and we do agree a concrete number would be beneficial. The area we reconstructed in experiment is approximately 1.2x1.2 mm². This has been added in the second last paragraph of section 3 of our main manuscript.

o As angle calibration is a critical component of APIC, I suggest to make it clear in the main text (now I can only find this discussion in the supplement) and reference the work that the calibration algorithm is based on.

Author response: We appreciate the reviewer's suggestion. However, we do want the readers to focus more on the idea of APIC and understand why one can obtain the closed-form solution for a complex field. These are the most important things and are the most novel aspects of APIC. Thus, we decided to place the implementation details in the supplementary. These details include the derivation of APIC and implementation details such as the number of NA-matching measurements for aberrations correction as well as the detailed calibration of the system. We have revised the manuscript to more clearly point the readers to the supplementary for these details.

C) Reference to related work

o The work has been mainly placed in the context of FPM. Due to the requirement of NA-matching measurement, it has been also placed in the context of a very recent (2021) work on space-domain Kramers–Kronig (KK) relation. I would like to make a few points:

§ The NA-matching requirement for efficient phase recovery based on intensity-only measurement was first pointed out by the work in 2019:

<https://www.spiedigitallibrary.org/journals/advanced-photonics/volume-1/issue-06/066004/High-speed-in-vitro-intensity-diffraction-tomography/10.1117/1.AP.1.6.066004.full?SSO=1>

- The 2019 work derives this critical requirement based on a transfer function (1st Born approximation) approach.

§ In a more recent work, the KK relation has been shown to be equivalent to the transfer function analysis (more precisely, an expanded 1st Rytov approximation model introduced in <https://www.nature.com/articles/s41377-022-00815-7>

Author response: We thank the reviewer for pointing this out. Our current version of APIC aims to image 2D thin sample. The first work shows that the NA-matching condition is important to phase reconstruction, which we completely agree. However, this point has been showed in other FPM works where the phase transfer function is derived. More importantly, the reason that NA-matching illumination should be used for Kramers-Kronig is more than that as the amplitude matters as well. As for the methodology, our method uses Kramers-Kronig method to perform reconstruction and the math for that is substantially different from the mentioned work.

With respect to the Rytov approximation, we want to gently point out that the Kramers-Kronig relation has a more general and relaxed assumption. In our derivation shown in the supplementary, we only require the unaltered field is stronger than the scattered field (Eq. S2), while 1st order Rytov assumes the variation of the complex refractive index is smooth and further ignores all the high order terms.

These references will likely be more relevant to potential 3D APIC imaging methods. As we are focused on 2D APIC imaging here, we feel that adding these references would just be confusing and distracting to general readers trying to understand APIC.

Minor comments:

- What is the initialization method used in the FPM reconstruction? Has the authors tried the differential phase contrast (DPC) based initialization method in Ref. 12 to alleviate some of the artifacts seen in the results?

Author response: DPC assumes that there is no aberration in the system while most of the measurements in our experiment were obtained from a system bearing different levels of aberration. We thus used the sum of the images as the initialization for FPM and this has been added in our main manuscript (second paragraph in Section 3).

- “Spatial-bandwidth product” – I believe usually it refers to as “Space-bandwidth product”.

Author response: We thank the reviewer for catching the typo. It has been corrected in our revised manuscript.

- Around Eqn S9, I suggest define $[NA/\lambda]$ to match the unit.

Author response: We thank the reviewer for pointing our this. We actually stated that NA is used to denote k_{na} for simplicity. We have re-emphasized this notation in the supplementary note.

REVIEWERS' COMMENTS

Reviewer #1 (Remarks to the Author):

Authors have addressed the minor questions raised and suggestions provided during the review. I recommend the manuscript for publication.

Reviewer #1 (Remarks on code availability):

Code was not available.

Reviewer #2 (Remarks to the Author):

This is a nice work. I support the publication of the work. The authors have addressed my concerns.

Reviewer #1 (Remarks to the Author):

Authors have addressed the minor questions raised and suggestions provided during the review. I recommend the manuscript for publication.

Author response: We thank for the feedback from the reviewer.

Reviewer #1 (Remarks on code availability):

Code was not available.

Author response: There was actually a statement that indicated the code for APIC was available on Github. We ordered the statement with respect to the code availability so it is easier to see now.

Reviewer #2 (Remarks to the Author):

This is a nice work. I support the publication of the work. The authors have addressed my concerns.

Author response: We thank the positive comment from the reviewer.